# Effects of individual base-pairs on in vivo target search and destruction kinetics of bacterial small RNA

Anustup Poddar [1], Muhammad S. Azam[2,6], Tunc Kayikcioglu[1], Maksym Bobrovskyy[2,7], Jichuan Zhang[1], Xiangqian Ma[2], Piyush Labhsetwar [3,8], Jingyi Fei [4], Digvijay Singh[1,9], Zaida Luthey-Schulten [3], Carin K. Vanderpool[2✉] & Taekjip Ha [1,5✉]

Base-pairing interactions mediate many intermolecular target recognition events. Even a single base-pair mismatch can cause a substantial difference in activity but how such changes influence the target search kinetics in vivo is unknown. Here, we use high-throughput sequencing and quantitative super-resolution imaging to probe the mutants of bacterial small RNA, SgrS, and their regulation of *ptsG* mRNA target. Mutations that disrupt binding of a chaperone protein, Hfq, and are distal to the mRNA annealing region still decrease the rate of target association, $k_{on}$, and increase the dissociation rate, $k_{off}$, showing that Hfq directly facilitates sRNA–mRNA annealing in vivo. Single base-pair mismatches in the annealing region reduce $k_{on}$ by 24–31% and increase $k_{off}$ by 14–25%, extending the time it takes to find and destroy the target by about a third. The effects of disrupting contiguous base-pairing are much more modest than that expected from thermodynamics, suggesting that Hfq buffers base-pair disruptions.

[1] Department of Biophysics and Biophysical Chemistry, Johns Hopkins University School of Medicine, Baltimore, MD, USA. [2] Department of Microbiology, University of Illinois at Urbana-Champaign, Urbana, IL, USA. [3] Department of Chemistry, University of Illinois at Urbana-Champaign, Urbana, IL, USA. [4] Department of Biochemistry and Molecular Biology, University of Chicago, Chicago, IL, USA. [5] Howard Hughes Medical Institute, Baltimore, MD, USA. [6] Present address: Department of Biochemistry and Molecular Biology, University of Chicago, Chicago, IL, USA. [7] Present address: Department of Microbiology, University of Chicago, Chicago, IL, USA. [8] Present address: The Land Institute, Salina, KS, USA. [9] Present address: Division of Biological Sciences, University of California, San Diego, CA, USA. ✉email: cvanderp@illinois.edu; tjha@jhu.edu

Myriad biological systems use base-pairing interactions for target recognition, where proteins mediate base-pairing interactions between two physically separated strands. Such base-pairing-mediated targeting is found in a wide range of processes, including DNA repair[1], noncoding RNA-based gene regulation[2,3], bacterial immunity using CRISPR[4], and therapies using antisense oligonucleotides[5]. They all rely on base-pairing interactions above a threshold for specificity. How do they achieve both accuracy and speed to sample through thousands of potential targets and rapidly reject nontargets? Recent advances in single-molecule imaging technologies made it possible to explore the kinetic parameters of target recognition and nontarget rejection in vitro[6–12], and in a limited number of cases, inside living cells[13]. However, we do not yet know the impact of single-nucleotide changes in in vivo target search kinetics, even though such minute changes can have large functional consequences. Our goal here is to quantify the mutational impact on base-pairing-mediated target search kinetics in vivo. We used bacterial gene regulation by small RNA (sRNA) as a model system.

Among the many examples of noncoding RNA-based gene regulation are microRNAs and long noncoding RNAs in eukaryotes, and sRNAs in bacteria and archaea[3,14,15]. Often, bacterial sRNAs regulate gene expression at a posttranscriptional level during stress, for example, in iron limitation stress[16], osmotic and acid stress[17], and oxidative stress[18]. Our work here studied the sRNA SgrS, which is produced in response to glucose-phosphate stress[19].

A disparity between sugar uptake and its metabolism gives rise to stress; a faster uptake leads to an accumulation of glucose-6-phosphate and activation of SgrR, a transcription factor. This stimulates the *sgrS* gene to transcribe SgrS, which reduces sugar transport, promotes efflux, and reroutes cellular metabolism[20–22]. Sugar stress conditions are provoked in most studies by subjecting cells to α-methylglucoside (αMG), a sugar analog that gets phosphorylated during import to form αMG-6-phosphate, which cannot be further processed metabolically. *Escherichia coli* SgrS, a 227-nt sRNA, binds reversibly and dynamically to its primary target, *ptsG* mRNA[23], which codes for the EIICB domain of the glucose phosphotransferase system. Binding between the RNAs, aided by a hexameric RNA chaperone protein Hfq, blocks the *ptsG* ribosome binding site, thereby inhibiting translation of new glucose transporters (Fig. 1a). This sRNA–mRNA complex also gets degraded by endoribonuclease RNase E, thus reducing the cellular concentration of *ptsG* mRNA. Hfq is important for the stability of sRNAs in general, and in vitro studies have shown that Hfq increases the rate of annealing between sRNA and its target mRNA sequences[24–26]. Whether Hfq also directly facilitates annealing between sRNA and mRNA in vivo is unknown for any sRNA, because it has not been possible to separate the effects of Hfq on sRNA stability and sRNA–mRNA annealing.

SgrS contains a 3′ Hfq-binding region predicted to contain two stem-loops, the small stem-loop, and the terminator stem-loop that is larger, followed by a U-rich tail (Fig. 1b)[27,28]. An optimal length of U-rich tail, with seven nucleotides or more[27,29], is required for the formation of functional sRNAs and for efficient Hfq binding, and Hfq binding to the two stem-loops is critical for target regulation[27,28,30–32].

Nucleotides 168–187 of SgrS are partially complementary to the *ptsG* 5′-UTR (Fig. 1c)[20]. Nucleotides 168–181, if presented as a 14 nt long oligonucleotide alone, are sufficient for full repression of *ptsG* translation in vitro and in vivo[33]. Among these, G176 and G178 have been shown to be most important for the annealing between SgrS and *ptsG* mRNA[24].

Previously, we developed a two-color 3D super-resolution imaging and modeling platform to determine in vivo target search kinetics for wild-type SgrS regulation of *ptsG*[34]. The bimolecular association rate constant $k_{on}$ between the RNAs was $2 \times 10^5$ M$^{-1}$ s$^{-1}$, which is within the wide range of reported Hfq-mediated sRNA and target mRNA association rates in vitro despite the crowded cellular environment and large excess of nontarget RNA molecules. The dissociation rate constant $k_{off}$ was 0.2 s$^{-1}$; 10–100-fold larger than in vitro estimates of other sRNA–mRNA pairs[32,35,36]. The large dissociation constant $K_D$ ($=k_{off}/k_{on}$) of ~1 μM explained why more than a hundred SgrS molecules are produced during *ptsG* mRNA regulation. The rate constant for co-degradation, $k_{cat}$, was surprisingly high, 0.4 s$^{-1}$, suggesting that RNA degradation machineries accompany the target search complex formed between SgrS and Hfq so that as soon as RNAs bind each other, RNAs can be degraded without waiting for the arrival of downstream degradation machineries. Finally, $k_{off}$ and $k_{cat}$ are similar in magnitude, suggesting that sRNA–mRNA complex is almost as likely to fall apart as to lead to co-degradation. Here, by expanding the scale of this quantitative imaging-based investigation by an order of magnitude to include ten SgrS mutants, we aimed to determine how $k_{on}$, $k_{off}$, and $k_{cat}$ are affected by single-nucleotide changes.

We formulated a pipeline of experiments to identify and examine the key regions in SgrS responsible for the annealing and regulation of *ptsG*. We used Sort-Seq, a high-throughput method that can estimate the impact of different mutations on the overall activity of the fluorescence reporter system chosen[37–39]. From the Sort-Seq results, we identified the regions in the SgrS sequence important for the overall regulation and chose nine single-nucleotide substitution mutants. *E. coli* strains containing these mutations or one double substitution mutation in their endogenous chromosomal copy were constructed and studied using single-molecule fluorescence in situ hybridization (smFISH) followed by two-color 3D super-resolution imaging, and modeling to determine $k_{on}$, $k_{off}$, and $k_{cat}$. Our results show that the two stem-loops at the 3′ end of SgrS play important roles in the activity of the sRNA. We also provide in vivo evidence that Hfq directly facilitates SgrS-*ptsG* mRNA base-pairing. Importantly, we were able to unambiguously ascribe relative contributions of single base-pairs to sRNA lifetimes and target search kinetics, allowing us to quantify by how much the rates of mRNA binding and rejection are influenced by eliminating a single base-pair between them.

## Results

### Sort-Seq reveals SgrS nucleotides important for target regulation.
We employed a high-throughput Sort-Seq approach to identify SgrS regions important for *ptsG* regulation. We created a low copy number reporter plasmid containing a partial *ptsG* sequence (105 nt 5′-UTR along with the first 30 nt coding sequence of *ptsG* mRNA) and superfolder GFP-coding sequence (*ptsG-sf*GFP)[40] (Supplementary Fig. 1), and transformed it into *E. coli* strain MB1 (Δ*ptsG*, Δ*sgrS*, *lacI*^q, and *tetR*). The *sgrS* mutation library was constructed by random mutagenesis PCR of a plasmid[40] containing the *sgrS* sequence (Supplementary Fig. 1) and was then transformed into the MB1 strain containing the reporter plasmid (Fig. 2a). The expression of *ptsG-sf*GFP and *sgrS* were under the control of P$_{Llac-O1}$ and P$_{Ltet-O1}$, respectively, and were induced by isopropyl β-D-1-thiogalactopyranoside (IPTG) and anhydrotetracycline (aTc). Upon induction by IPTG, cells containing the target reporter (*ptsG-sf*GFP) alone showed bright fluorescence, while those co-transformed with the plasmid containing wild-type *sgrS* showed weak GFP fluorescence in the presence of both IPTG and aTc in single-cell imaging (Supplementary Fig. 2) and flow cytometry analysis (Fig. 2b and Supplementary Fig. 54), indicating an effective repression of the reporter. Cells co-transformed with the *sgrS* mutant library

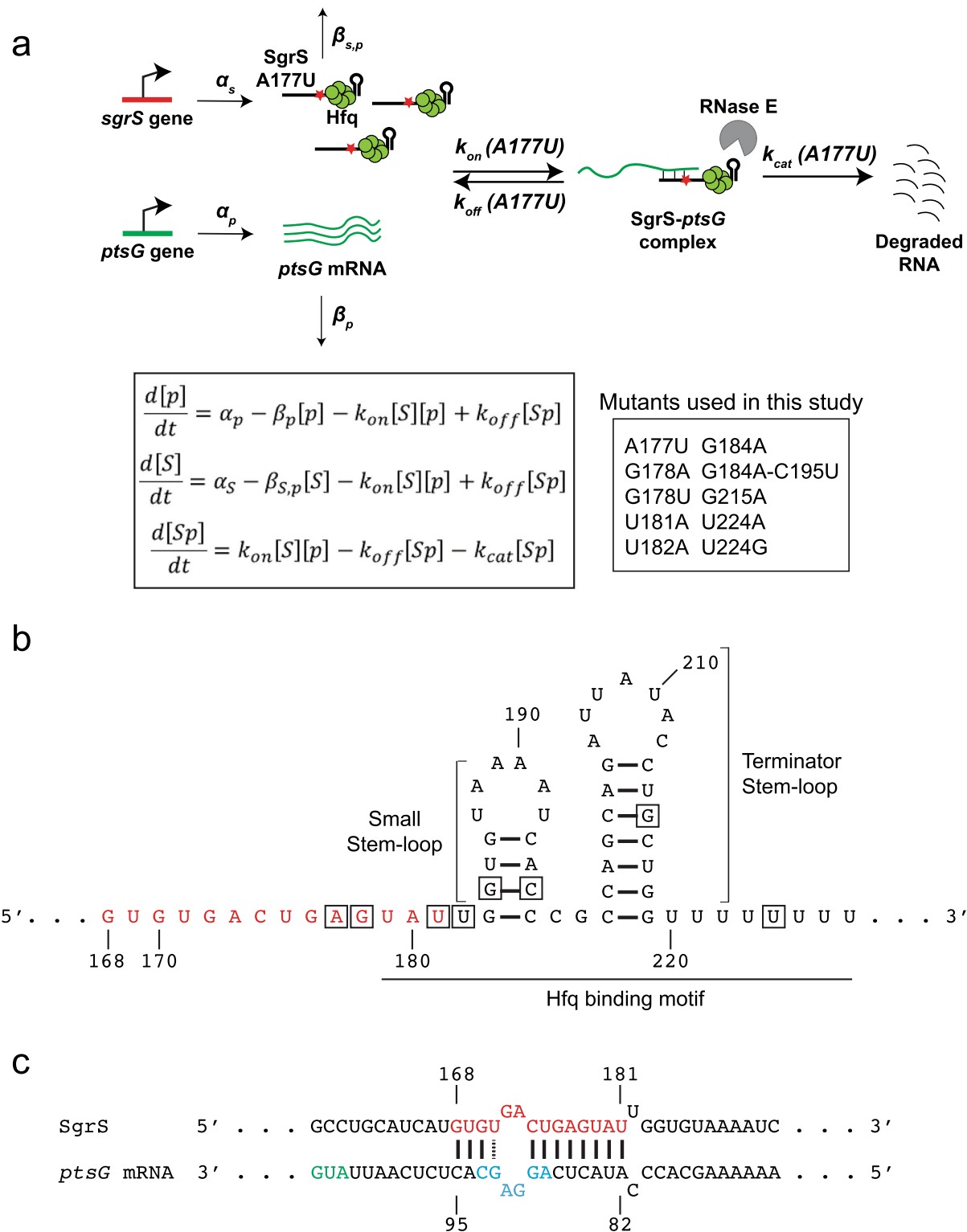

**Fig. 1 Target search kinetics of SgrS. a** Kinetic scheme of *ptsG* mRNA degradation induced by wild-type SgrS sRNA and the different SgrS point mutants. The figure shows one of the SgrS mutant strains, A177U. The red star represents the position of the mutation from A177 to U. The steps are described in detail in the main text and the inset shows the mutants used in this study. [*p*], [*S*], and [*Sp*] are the concentrations of *ptsG* mRNA, SgrS, and the SgrS-*ptsG* complex, respectively, in their mass-action equations. The *α*'s are the rates of transcription and *β*'s are the rates of degradation of the RNAs; $k_{on}$, $k_{off}$, and $k_{cat}$ are the rates of association, dissociation, and co-degradation, respectively. **b** Secondary structure of SgrS sRNA from nucleotide 168 to the poly-U tail. The nucleotide positions where mutations were made are boxed. The nucleotides involved in base-pairing with *ptsG* mRNA are red. **c** Base-pairing interaction between SgrS and *ptsG* mRNA showing the complementary region, start site, and ribosome binding site.

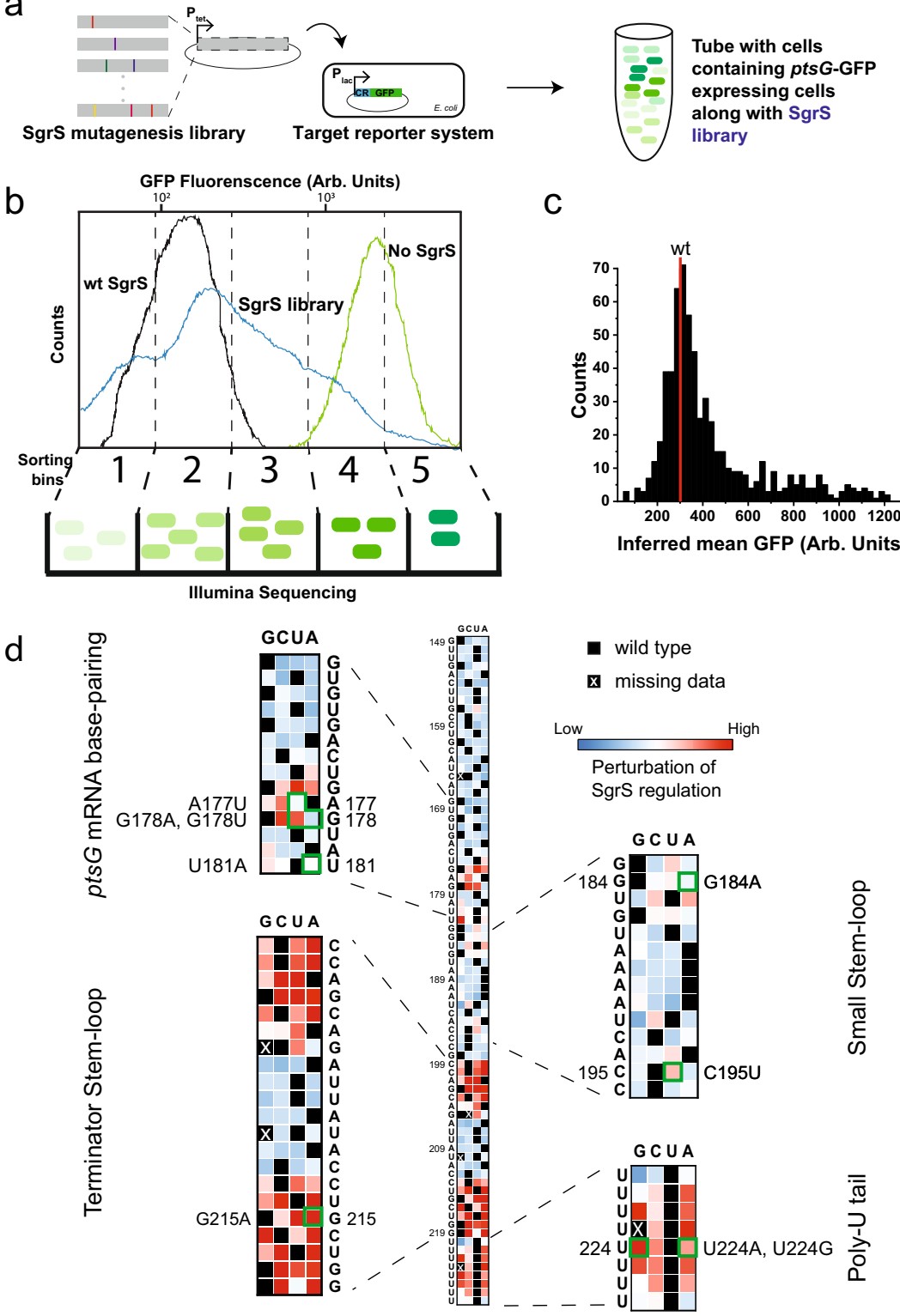

showed a broad distribution of GFP fluorescence indicating highly variable levels of regulation by mutants (Fig. 2b). Based on the flow cytometry results, the cells were collected in five intensity bins, and the plasmids were extracted from the cells. For each bin, the mutated *sgrS* sequence from position 149 to 227 was amplified by PCR and sequenced. Sequencing was limited to this region because the 5′ region, up to nucleotide 153 and coding for the 43

amino acid peptide SgrT, is not involved in base-pairing-dependent mRNA regulation[41,42].

Using the relative abundance of sequences in each bin and the GFP fluorescence levels from the flow cytometry analysis, we calculated, for each single point mutation, the average fluorescence intensity of cells sharing the same mutation as a measure of the regulation defect (Fig. 2d)[39]. High average single-cell

**Fig. 2 Mapping efficacy of SgrS regulation of ptsG mRNA with respect to its sequence. a** Preparation of SgrS mutation library. Mutations were introduced into the SgrS plasmid using mutagenesis PCR. The mutations introduced are represented by the colored bars. This library was then transformed into an *E. coli* strain already transformed with the *ptsG* mRNA plasmid fused with a GFP reporter. The varying levels of GFP fluorescence are shown by the green colors. **b** Sorting of the cells and sequencing. The cells with two-plasmid co-expression were sorted using flow cytometry. The SgrS library (blue) shows GFP fluorescence that spans the region from the wild-type SgrS (black) to target (*ptsG*)-only (green). Cells were sorted into five evenly spaced (log scale) fluorescence bins and the occupancy percentages were 18.74%, 33.76%, 30.91%, 13.83%, and 2.76%, respectively. The cells from each bin were grown, DNA was purified, barcoded, and sequenced using Illumina sequencing platform. **c** Histogram of the Sort-Seq measurements from the SgrS library from two replicates combined. The mean fluorescence for the wild-type SgrS is shown in red. **d** Heatmap showing the effect of mutations on SgrS regulation of *ptsG* mRNA starting from nucleotide 149 to 227. The colors in the boxes are scaled from blue (low) to red (high), according to the level of perturbation of SgrS regulation. Black squares represent the wild-type base at each position and the black boxes with white crosses show the positions of the mutants missing in the experiment. Text shows the wild-type sequence of SgrS. Insets show the four regions of SgrS, viz. base-pairing region, small stem-loop, terminator stem-loop, and the poly-U tail. Source data are provided as a Source data file.

fluorescence would correspond to SgrS mutants that are highly defective in regulation of *ptsG* reporter expression and vice versa. The degree of perturbation to the regulatory capacity is color-coded in the heat map grid, ranging from the least (blue) through intermediate (white) to the most (red). Nucleotides 149–174 showed little to no perturbation of SgrS regulation as shown by the blue squares (Fig. 2d). In contrast, the region where SgrS can base-pair with *ptsG* mRNA (U175 to G186) displayed perturbations across a wide range, as shown by the white and red squares in the grid. Specifically, previous studies showed that G176C or G178C eliminates the SgrS's ability to downregulate *ptsG*, while C174G and G170C only weakly perturb SgrS function in vivo and in vitro[24,31]. The corresponding squares in our heatmap grid (Fig. 2d) show red or dark red for G176C and G178C, and white or light blue for C174G and G170C, validating our Sort-Seq results. We also see that SgrS regulation is hampered if there are mutations in the small stem-loop region (nts 183–196 (Fig. 1b)), the terminator stem-loop region (nts 199–219), and the poly-U tail (nts 220–227). The largest effect is seen in the stem region of the terminator stem-loop, C199 to G205, and C213 to G219, where we see the darkest red, highlighting the importance of this stem-loop. These stem-loop regions and the poly-U tail play a role in Hfq binding[27,28], and our Sort-Seq analysis therefore confirms that Hfq interaction is important for SgrS function in the cell.

Based on Sort-Seq results, we picked nine single point mutations for further investigation. These include mutations in the target-annealing region (A177U, G178A, G178U, and U181A), U-rich region upstream of the small stem-loop (U182A), the small stem-loop (G184A), the terminator stem-loop (G215A), and the poly-U tail (U224G and U224A).

**SgrS mutation effects on regulation of *ptsG* reporter**. To examine the effect of the selected SgrS mutations on *ptsG* regulation, we monitored the effect of wild-type and seven of the SgrS mutants (plasmid-encoded and expressed from an inducible promoter) on the activity of a chromosomal *ptsG'-'lacZ* translational fusion (Fig. 3a). The wild-type SgrS almost completely eliminated β-galactosidase activity, whereas the mutants showed regulation defects of various degrees consistent with the Sort-Seq data. SgrS G215A, which disrupts the terminator stem-loop structure, showed the largest defect. To test if the regulatory defects can be explained by a reduction of SgrS levels, for example, due to shorter cellular lifetimes associated with impaired Hfq binding, we performed northern blot analysis. We found that SgrS abundance is not affected for four of the mutants (A177U, G178U, G178A, and G184A) and is reduced by 40–50% for mutations in the terminator stem-loop or poly-U tail (G215A, U224G, and U224A; Fig. 3c). Interestingly, the latter three mutants showed large increases in readthrough transcription, suggesting that transcription termination is defective (Fig. 3d). These observations are consistent with a previous study which

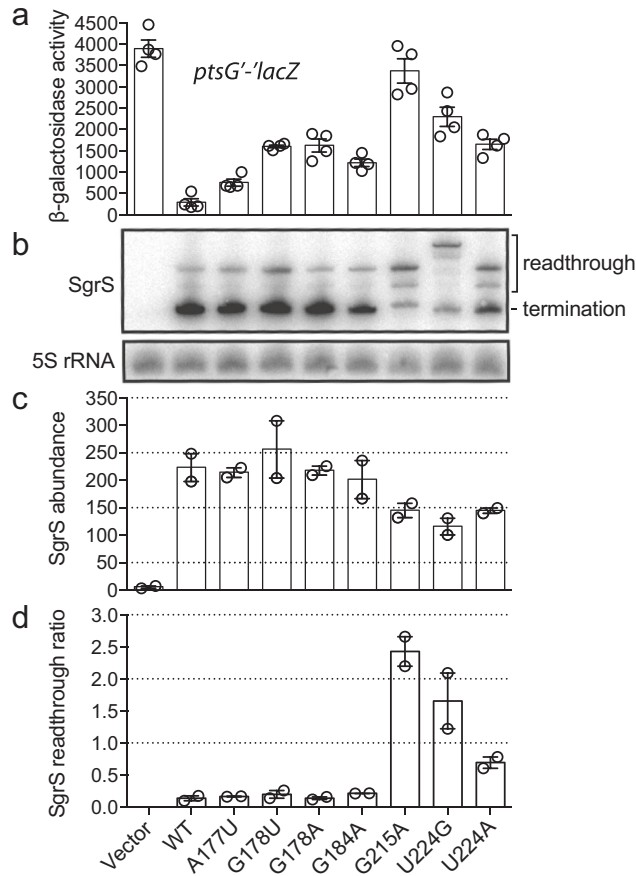

**Fig. 3 Regulation of *ptsG'-'lacZ* translational fusion by SgrS point mutants. a** Regulation of chromosomal *ptsG'-'lacZ* translational fusion by wild-type SgrS and A177U, G178U, G178A, G184A, G215A, U224A, and U224G mutant variants (plasmid-encoded) was assessed using β-galactosidase activity assay. Data were obtained from $n = 4$ independent experiments and presented as mean ± SEM. **b** RNA was extracted simultaneously with β-galactosidase activity assay and northern blot was performed on two biological replicates using probes specific for SgrS sRNA and 5S rRNA (control). Full-length (227 nt), properly terminated SgrS transcripts are labeled as "termination" products, and longer transcripts that arose due to transcriptional readthrough are labeled as "readthrough" products. **c** Band intensities of total SgrS transcripts (termination + readthrough) were measured, and 5S-normalized values were plotted as "SgrS abundance" (steady-state transcript abundance of SgrS mutants). Data were obtained from $n = 2$ independent experiments and presented as mean ± SEM. **d** Band intensities of SgrS termination and readthrough products were measured and 5S-normalized ratios (readthrough/termination) were calculated and plotted for each SgrS mutant as "SgrS readthrough ratio". Data were obtained from $n = 2$ independent experiments and presented as mean ± SEM. Source data are provided as a Source data file.

showed that SgrS molecules with an extended 3′ region do not interact with Hfq[43]. Overall, single point mutations outside the large terminator stem-loop and poly-U tail have minimal impact on SgrS abundance, and their regulatory defects cannot be explained by SgrS abundance changes.

**Super-resolution imaging of specific chromosomal SgrS mutants.** A set of nine single point mutants of SgrS were chosen for further analysis using quantitative imaging (A177U, G178A, G178U, G184A, U181A, U182A, G215A, U224A, and U224G). To avoid potential complications arising from SgrS overexpression, we created these mutations in the endogenous chromosomal copy of SgrS. A177, G178, and U181 are in the seed (target base-pairing) region, G184 is in the small stem-loop region. U181 and U182 are in the U-rich region upstream of the small stem-loop, previously shown to bind Hfq[27]. We also constructed a double mutant, G184A–C195U, which restores the small stem-loop structure. G215 is in the terminator stem-loop region and U224 is in the poly-U tail, both of which provide major binding sites for Hfq. These mutant alleles in the background of strains with wild-type RNase E or a C-terminally truncated RNase E were grown, and glucose-phosphate stress was induced using αMG for a varied amount of time before cell fixation and permeabilization. We performed two-color 3D super-resolution imaging of the SgrS sRNAs labeled with up to 9 FISH probes conjugated to Alexa Fluor 647 and the ptsG mRNAs labeled with up to 28 FISH probes conjugated to CF 568. ΔsgrS and ΔptsG strains were also examined to correct for the background arising from nonspecific binding of FISH probes. The wild-type strain showed an increase in SgrS copy number over time after sugar stress induction (Fig. 4 and Supplementary Fig. 3). At the same time, the copy number of ptsG mRNA showed a decrease (Fig. 4 and Supplementary Fig. 3). We used a density-based clustering algorithm[34] to determine the copy numbers of RNAs along with the copy number of SgrS-ptsG mRNA complexes. Super-resolution imaging was especially important for quantifying sRNA–mRNA complexes because at conventional microscopy resolution there was too much false colocalization between sRNA and mRNA.

The total copy number of the mutant SgrS sRNAs was lower than for wild-type SgrS with an accompanying impairment in ptsG mRNA degradation for all single point mutants examined, showing that their regulatory functions are perturbed (Fig. 4, and Supplementary Figs. 4–9 and 11–13). The single-cell distribution of RNA copy numbers also showed a decreased copy number of SgrS, with the histogram peaking at lower copy numbers 20 min after αMG induction, and the histograms for ptsG mRNA peaked at higher copy numbers per cell compared to the wild type (Fig. 5b, d, h and Supplementary Fig. 52). The lowest copy number of SgrS was seen for G184A and G215A, and they also showed the most impaired mRNA degradation (Figs. 4 and 5a, c, g, and Supplementary Figs. 9 and 11). These two mutations occur in two separate stem-loop regions, both of which participate in Hfq binding[27,28]. The double mutant, G184A–C195U, which restores base-pairing in the small stem-loop via a compensatory mutation, eliminated the negative impact of G184A as seen by recovery of SgrS accumulation and regulation of ptsG mRNA (Figs. 4 and 5e, f, and Supplementary Fig. 10). This suggests that the disruption of the stem-loop structure, not of G184 base-pairing with ptsG mRNA, is primarily responsible for regulatory defects of G184A.

These imaging data by themselves cannot tell us whether regulatory defects are due to changes in target binding kinetics or due to changes in the SgrS stability. Therefore, we next determined the lifetimes of wild-type and mutant SgrS molecules.

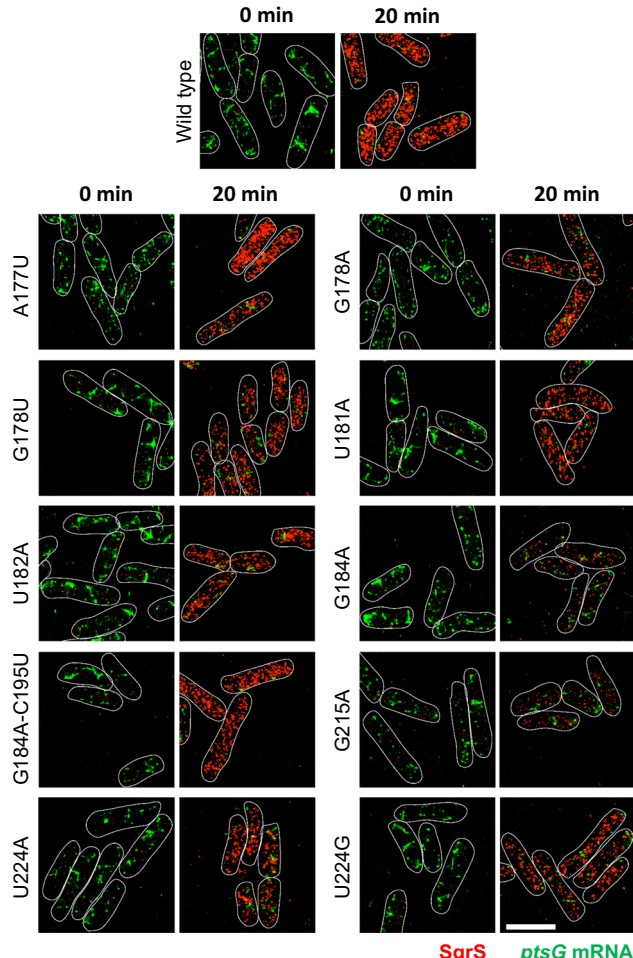

**Fig. 4 3D super-resolution images projected in 2D planes.** The panels show SgrS (red) and ptsG mRNA (green) labeled by smFISH for the wild-type and the mutant strains, A177U, G178A, G178U, U181A, U182A, G184A, G184A–C195U, G215A, U224A, and U224G before and after 20 min of αMG (non-metabolizable sugar analog) induction. Each experiment was performed independently two times. Cell boundaries are denoted by white solid lines. Scale bar is 2 μm and applies to all the panels.

**Intrinsic lifetimes of SgrS mutants.** In order to calculate the target-independent lifetime of SgrS, we induced SgrS expression using αMG and then added rifampicin to stop transcription globally. RT-qPCR was performed vs time after rifampicin treatment to quantify the SgrS level. The wild-type SgrS showed minimal intrinsic degradation over a period of 2 h after the addition of rifampicin, but it showed rapid degradation in the presence of ongoing transcription (10.4 ± 0.7 min), suggesting that SgrS degradation is normally dominated by co-degradation with its various target mRNAs (Supplementary Fig. 25). The intrinsic degradation was also minimal for SgrS A177U mutant (Supplementary Fig. 25), suggesting that in the absence of co-degradation, a mutation in the target-annealing region does not destabilize SgrS. In contrast, intrinsic degradation of G184A was rapid (lifetime of 6.3 min) and so was the intrinsic degradation of wild-type SgrS in Δhfq strain (lifetime of 5.1 min; Supplementary Fig. 25), indicating that Hfq is required for the target-independent stability of SgrS and the small stem-loop is important for Hfq binding.

**Lifetime of SgrS mutants.** In order to determine the effective lifetime of SgrS mutants, which includes the contributions from

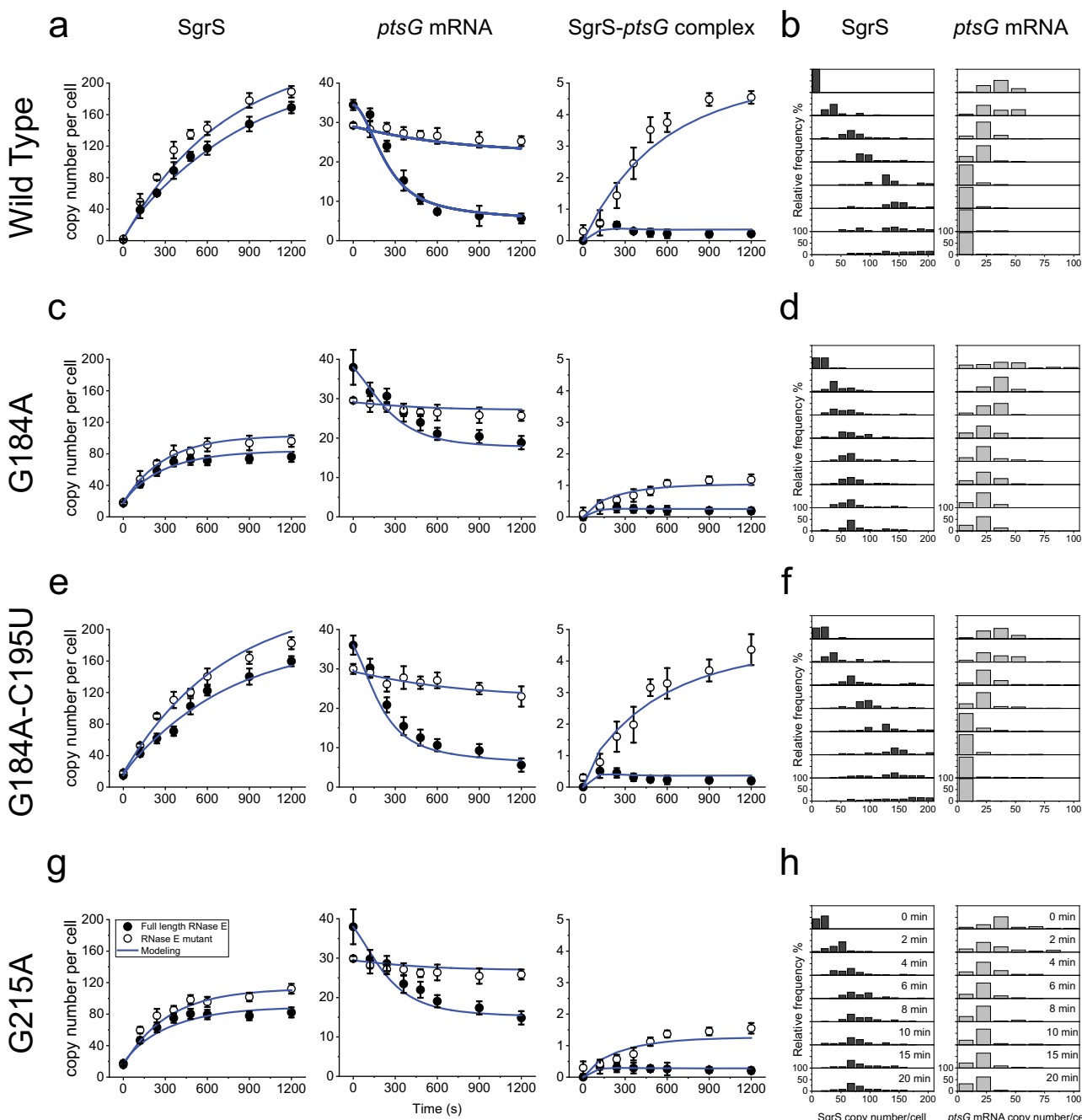

**Fig. 5 Time-dependent changes in the copy numbers of SgrS and *ptsG* mRNA and estimation of kinetic parameters.** Time course changes and corresponding modeling curves for the SgrS, *ptsG* mRNA, and SgrS-*ptsG* complex in **a** wild type, **c** G184A mutant strain, **e** G184A–C195U mutant strain, **g** G215A mutant strain. Average copy numbers per cell are plotted against time. Rate constants obtained for these mutants are shown in Fig. 6 and in Supplementary Table 3. Weighted $R^2$'s for modeling are also reported in Supplementary Table 4. Data are presented as mean values ± SEM for $n = 98$, 90, 169, 144, 149, 127, 94, and 82 cells examined over two independent experiments after 0, 2, 4, 6, 8, 10, 15, and 20 min induction respectively for **a**; for $n = $ 84, 84, 94, 99, 94, 90, 87, and 88 cells examined over two independent experiments after 0, 2, 4, 6, 8, 10, 15, and 20 min induction, respectively, for **c**; for $n = 117$, 101, 115, 111, 102, 110, 104, and 104 cells examined over two independent experiments after 0, 2, 4, 6, 8, 10, 15, and 20 min induction, respectively, for **e** and for $n = 82$, 88, 94, 104, 94, 90, 89, and 88 cells examined over two independent experiments after 0, 2, 4, 6, 8, 10, 15, and 20 min induction, respectively, for **g**. Histograms showing the change in distribution of SgrS and *ptsG* mRNA copy numbers for **b** wild type for $n = 98$, 90, 169, 144, 149, 127, 94, and 82 cells examined over two independent experiments after 0, 2, 4, 6, 8, 10, 15, and 20 min induction, respectively, **d** G184A mutant strain for $n = $ 84, 84, 94, 99, 94, 90, 87, and 88 cells examined over two independent experiments after 0, 2, 4, 6, 8, 10, 15, and 20 min induction, respectively, **f** G184A–C195U mutant strain for $n = 117$, 101, 115, 111, 102, 110, 104, and 104 cells examined over two independent experiments after 0, 2, 4, 6, 8, 10, 15, and 20 min induction, respectively, **h** G215A mutant strain for $n = 82$, 88, 94, 104, 94, 90, 89, and 88 cells examined over two independent experiments after 0, 2, 4, 6, 8, 10, 15, and 20 min induction, respectively. Source data are provided as a Source data file.

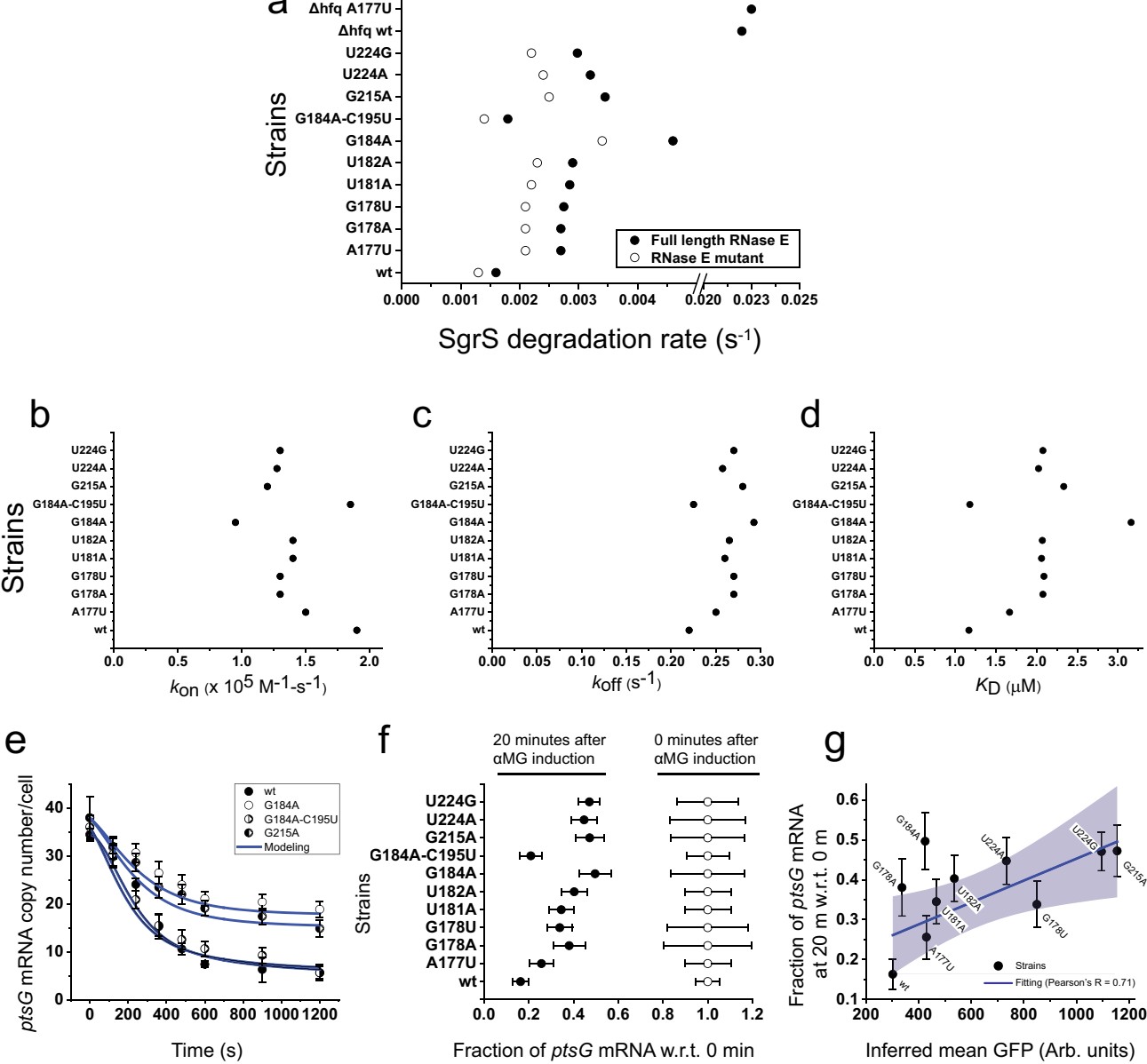

intrinsic degradation and co-degradation with target mRNA, the strains carrying chromosomal mutations were treated with αMG for 10 min before rinsing it away. SgrS decay over time was then monitored through imaging of fixed cells. The wild-type SgrS showed a degradation rate of $0.0016\,s^{-1}$ (lifetime of 10.4 min) and all of the mutants showed higher rates, the highest being for G184A with $0.0046\,s^{-1}$ (lifetime of 3.6 min), followed by G215A with $0.00345\,s^{-1}$ (lifetime of 4.8 min; Fig. 6a, and Supplementary Figs. 26–32 and 34–36). Because G184A and G215A disrupt the small and terminator stem-loop regions, respectively, our data suggest that both stem-loop regions are important for SgrS stability in vivo. G184A–C195U recovered the stability of SgrS to the wild-type level with an identical degradation rate within error (Fig. 6a and Supplementary Fig. 33). Rifampicin-chase experiments did not show any difference in the lifetime of *ptsG* mRNA between all mutant strains (Supplementary Note 1, and Supplementary Figs. 26–37), showing that the mutations in SgrS have no effect on *ptsG* mRNA stability when SgrS is not induced.

The degradation rate of SgrS in Δ*hfq* strains was much higher, $\sim0.022\,s^{-1}$ (lifetime of 0.76 min), for wild-type and all SgrS mutants

(Fig. 6a and Supplementary Fig. 38–40)[23]. This 14-fold increase in degradation rate for sRNAs in Δ*hfq* strains confirms that Hfq is indispensable for the stability of SgrS[27,44]. Because none of the SgrS mutants in the *hfq*+ cells showed degradation rates as high as in Δ*hfq* strains, these SgrS mutations are only partially deleterious to the interactions with Hfq.

Mutations in the base-pairing regions (A177U, G178U, and G178A) reduced the lifetime of SgrS in the imaging-based experiment even though they are not expected to alter Hfq binding. Because our northern blot analysis of overexpressed SgrS showed that these mutations do not change SgrS abundance, the intrinsic degradation is unlikely to be affected by the mutations. Instead, we attribute the discrepancy to mutation-induced alterations in co-degradation of SgrS with other SgrS target mRNAs.

**Target search and destruction kinetics of SgrS mutants**. Once we obtained the average copy numbers of SgrS, *ptsG* mRNA, and the SgrS-*ptsG* complex per cell as a function of time after SgrS

**Fig. 6 Calculation of various parameters and correlation with Sort-Seq. a** Degradation rates of SgrS for the wild type and the strains A177U, G178A, G178U, U181A, U182A, G184A, G184A–C195U, G215A, U224A, U224G, Δ*hfq* wild type, Δ*hfq* A177U, and Δ*hfq* G184A for full-length RNase E and RNase E mutants. Error bars represent standard deviation from two experimental replicates. **b–d** $k_{on}$, $k_{off}$, and $K_D$ measured from the time-dependent modeling curves of the SgrS, *ptsG* mRNA, and SgrS-*ptsG* mRNA complexes for the wild type and strains A177U, G178A, G178U, U181A, U182A, G184A, G184A–C195U, G215A, U224A, and U224G. These were determined simultaneously in the wild-type and RNase E mutants. Error bars report standard deviation from the independent fitting on two replicates. **e** Time course changes in *ptsG* mRNA for the wild type, G184A, G184A–C195U, and G215A mutant strains. Data are presented as mean values ± SEM from $n = 98, 90, 169, 144, 149, 127, 94$, and 82 cells examined over two independent experiments after 0, 2, 4, 6, 8, 10, 15, and 20 min induction, respectively, for wild type; for $n = 84, 84, 94, 99, 94, 90, 87$, and 88 cells examined over two independent experiments after 0, 2, 4, 6, 8, 10, 15, and 20 min induction, respectively, for G184A mutant strain; for $n = 117, 101, 115, 111, 102, 110, 104$, and 104 cells examined over two independent experiments after 0, 2, 4, 6, 8, 10, 15, and 20 min induction, respectively, for G184A–C195U mutant strain and for $n = 82, 88, 94, 104, 94, 90, 89$, and 88 cells examined over two independent experiments after 0, 2, 4, 6, 8, 10, 15, and 20 min induction, respectively, for G215A mutant strain. **f** Fractional change in *ptsG* mRNA copy numbers for the wild type and the mutants A177U, G178A, G178U, U181A, U182A, G184A, G184A–C195U, G215A, U224A, and U224G before and after 20 min αMG induction. Data are presented as mean values ± SEM from $n = 98$ and 82 cells examined over two independent experiments after 0 and 20 min induction for wild type; $n = 132$ and 80 cells examined over two independent experiments after 0 and 20 min induction for A177U mutant strain; $n = 81$ and 82 cells examined over two independent experiments after 0 and 20 min induction for G178A mutant strain; $n = 88$ and 83 cells examined over two independent experiments after 0 and 20 min induction for G178U mutant strain; $n = 91$ and 97 cells examined over two independent experiments after 0 and 20 min induction for U181A mutant strain; $n = 101$ and 94 cells examined over two independent experiments after 0 and 20 min induction for U182A mutant strain; $n = 84$ and 88 cells examined over two independent experiments after 0 and 20 min induction for G184A mutant strain; $n = 117$ and 104 cells examined over two independent experiments after 0 and 20 min induction for G184A–C195U mutant strain; $n = 82$ and 88 cells examined over two independent experiments after 0 and 20 min induction for G215A mutant strain; $n = 90$ and 84 cells examined over two independent experiments after 0 and 20 min induction for U224A mutant strain; $n = 92$ and 100 cells examined over two independent experiments after 0 and 20 min induction for U224G mutant strain. **g** Comparison of the SgrS regulation efficacy calculated from Sort-Seq assay and the imaging-based analysis. Error bars in the *x*-axis are standard deviations calculated from two experimental replicates and those in the *y*-axis are as described in **f**. The fitting is shown in blue and the gray region shows the 95% confidence region. Pearson's $R = 0.71$; 95% CI = 0.39, 0.88; $P = 0.021$, two-sided *t* test. Source data are provided as a Source data file.

induction for the full-length RNase E and RNase E mutant cases (Supplementary Figs. 14–24), we used a previously developed deterministic kinetic model to describe the SgrS-*ptsG* regulation kinetics (Fig. 1a)[34]. We used the experimentally determined degradation rate for *ptsG* mRNA, $\beta_p$, to calculate the *ptsG* transcription rate $\alpha_p$ using $\alpha_p = \beta_p \times [p]_0$, where $[p]_0$ is the steady-state copy number of *ptsG* mRNA at $t = 0$. By globally fitting the six time courses of the three RNA species with or without RNase E mutation that inhibits co-degradation, we obtained $k_{on}$, $k_{off}$, and $k_{cat}$ for the wild-type and mutant SgrS.

$k_{on}$ for the wild-type strain was $1.9 \times 10^5\,M^{-1}\,s^{-1}$ and $k_{off}$ was $0.22\,s^{-1}$ giving a $K_D$ of $1.16\,\mu M$, comparable to the previously published results (Fig. 6b–d)[34]. $k_{on}$ was lower for all single point mutants compared to the wild type and the reduction ranged from 24% for A177U to 53% for G184A. $k_{off}$ was higher for all the mutants and the increase ranged from 14% for A177U to 33% for G184A giving a dissociation constant, $K_D$ of 1.67 and 3.08 μM, respectively (Fig. 6b–d). Rate of transcription of SgrS and $k_{cat}$ were not affected by the mutations within error (Supplementary Figs. 42 and 43). To test the possibility that the apparent changes in $k_{on}$ and $k_{off}$ are due to fitting errors, and that the regulatory deficiencies can be explained solely by reduction in SgrS lifetimes, we repeated the global fitting procedure while keeping the $k_{on}$ and $k_{off}$ values fixed at the wild-type values. The fits were considerably worse, and were especially poor for copy number curves of *ptsG* mRNA and SgrS/mRNA complex (Supplementary Figs. 44–51). Therefore, our procedure of obtaining the mutation effects on $k_{on}$ and $k_{off}$ is robust.

When we restored the base-pairing in the small stem-loop by adding a compensatory mutation to G184A, the mutant that showed the largest changes to $k_{on}$ and $k_{off}$ (G184A–C195U), $k_{on}$ and $k_{off}$ returned to the wild-type values within error (Fig. 6b–d). Nucleotides 168–187 in SgrS were originally proposed to participate in base-pairing with the *ptsG* mRNA[20], but a subsequent study showed that only nucleotides 168–181 are required for base-pairing[33]. Here, we found that binding kinetics are similar between the wild type and G184C–C195U, strongly suggesting that a mutation at G184 primarily acts through

disruption of the small stem-loop structure, thereby affecting Hfq binding, instead of through direct disruption of base-pairing of G184 with the target strand. Even though our data suggest that G184 is not involved with SgrS-*ptsG* mRNA base-pairing, its mutation negatively affected annealing kinetics, decreasing $k_{on}$ and increasing $k_{off}$. Therefore, our results support the dual roles of Hfq: first to increase sRNA stability (Fig. 6a) and second to directly facilitate SgrS-*ptsG* binding.

The A to U mutation at position 177 removes an AU base-pair, breaking eight base-pairs, the longest stretch of contiguous base-pairing between SgrS and *ptsG* mRNA into segments of four and three base-pairs. This disruption gives a reduction in association rate of 24% and an increase in dissociation rate by 14%. The two mutations at position 178 eliminate a GC base-pair, and breaking the same eight base-pairs into segments of three and four base-pairs. G178A and G178U mutants gave a reduction of association rate by 31–32% and an increase of dissociation rate by 23–25%. The larger effects of G178 mutations compared to A177U are likely due to the loss of GC over AU base-pair. Consistent with this suggestion, a mutation at U181, losing an AU base-pair, decreased the association rate by 26% and increased the dissociation rate by 18%, very similar to A177U values.

The G215A mutation in the terminator stem-loop and the mutations U224A and U224G in the poly-U tail showed $k_{on}$ decreases and $k_{off}$ increases even though they should not change complementarity between SgrS and *ptsG* mRNA. The substantial effects on binding kinetics must therefore be due to defects in Hfq's ability to facilitate the annealing reaction, further providing in vivo evidence of direct facilitation of base-pairing between sRNA and mRNA by Hfq.

**Difference in regulation outcome between imaging and Sort-Seq experiments.** To examine if the regulation outcomes for SgrS mutants are consistent between our quantitative imaging experiments and Sort-Seq analysis, we used the fractional decrease of *ptsG* mRNA over the first 20 min after sugar stress induction as a measure of the SgrS regulation of *ptsG* mRNA target in imaging-based analysis. (Fig. 6e–g). Plotting these values

vs the inferred GFP signals obtained from the Sort-Seq experiments, we observed a relatively weak correlation (Pearson's $R = 0.71$), suggesting that the translation reporter-based Sort-Seq method is not able to fully capture the regulation defects of SgrS mutations. For example, G178A which had a large deficiency in regulation in imaging experiments showed almost the wild-type level regulation in Sort-Seq. A large defect in regulation was shown in previous studies where nucleotide 178 was mutated and our imaging-based approach is in accordance with this finding[24]. SgrS overexpression in Sort-Seq may have overcome the negative effect of mutations through mass action when the defect is primarily in binding kinetics. For the G215A, U224A, and U224G mutant strains, however, Sort-Seq showed large regulatory deficiencies, suggesting that their defects cannot be overcome by overexpression. There are several possible explanations. First, because Sort-Seq relies on the translational output, mutations that disrupt translational inhibition but not RNA co-degradation may not be scored well in imaging-based experiments. However, in vitro translation experiments showed that mutation of G178 to C eliminates translation inhibition by SgrS[33], making it unlikely that defects in RNA–RNA annealing do not affect translational repression. Second, even when these mutants can bind Hfq, the complex may be defective in mediating RNA annealing. Third, these mutations also interfere with proper termination as shown by readthrough transcripts (Fig. 3). It was shown previously that the readthrough products of SgrS transcription do not bind Hfq in vivo and in vitro[43]. Weakening of the terminator stem-loop or reduction of the slippery Us must be causing transcription readthroughs that produce regulation-defective products, and much of the effect of G215A, U224A, or U224G may be due to improper termination. It has also been shown that readthrough transcription of SgrS is suppressed under stress conditions, providing an additional layer of regulation[43]. Finally, the incorporation of the GFP in the reporter system may have affected the stability of the mRNA.

## Discussion

Previous studies have measured the effect of mutations in the regulation of mRNA targets of SgrS[45–47] and have shown the importance of Hfq, sequence complementarity between SgrS and its targets, and RNA secondary structures[27,33,48,49]. Hfq has also been shown to promote structural changes to the RNAs, which in turn helps in the annealing and, consequently, regulation[50–52]. Our study provides a quantitative description of the process of target search and off-target rejection by determining the kinetic parameters as a function of single-nucleotide changes in functionally important regions. The $k_{on}$ and $k_{off}$ values determined in this study depict the apparent rate constants because we did not explicitly include Hfq binding in our model. $k_{on}$ in particular should have contributions from Hfq binding to SgrS, target search by SgrS/Hfq and subsequent annealing.

We used IntaRNA[53–56] to predict the energy of interaction between SgrS and ptsG mRNA, and found that it changes by ~6.4 kcal mol$^{-1}$ for the G178 point mutations, whereas the change is only ~4.3 kcal mol$^{-1}$ for A177U. Our study agreed with the ranking because we saw lower rates of association and higher rates of dissociation for G178 than A177U. However, the magnitude of the effect is much more modest compared to a simple prediction based on the energetic penalty. Both mutations introduce a mismatch within eight contiguous base-pairs, incurring large energetic penalties. For example, 6.4 kcal mol$^{-1}$ would correspond to a change in the equilibrium binding constant by a factor of ~60,000 instead of ~2 as we observed. Therefore, Hfq must be buffering the effect of breaking internal base-pairs in short helices. How this is achieved is presently unknown.

We found that the rate of co-degradation remains high, ~0.3 s$^{-1}$, even with SgrS mutations. The co-degradation of the SgrS-ptsG complex is brought about by the degradosome, in which RNase E is a key component. Hfq copurifies with RNase E and SgrS[57], and at least one sRNA (MicC) has been shown to mediate the interaction between Hfq and the C-terminal part of RNase E in vitro[58] and in vivo[59]. It has been hypothesized that the sRNA-Hfq-RNase E complex forms first and subsequently the complementary mRNA binds to this complex, aided by Hfq, followed by a coupled or sequential degradation of the RNA pair[58]. If so, once a stable complex with all four components forms, the co-degradation occurs at the same rate irrespective of the SgrS mutations as we observed.

Because $k_{cat}$ did not change with SgrS mutations, the probability that a single-binding event will cause co-degradation of sRNA and mRNA decreases with a mutation-induced increase in $k_{off}$. On average, the wild-type SgrS would take 1.73 $(=(k_{off} + k_{cat})/k_{cat})$ binding events before co-degradation. At the full SgrS accumulation condition, 100–200 copies per cell, corresponding to 0.48 µM (assuming 0.7 µm$^3$ per cell), the wild-type SgrS would take ~11 s to bind ptsG mRNA and the overall time it takes to degrade the target would be 19 s. The altered $k_{on}$ and $k_{off}$ values of the single point mutants would extend this target search and destruction time by about a third. This is a modest effect but for sRNAs that target multiple genes, single point mutations in sRNA or mRNA targets may alter the hierarchy of target regulation.

The poly-U sequence at the 3' end of sRNAs is an important Hfq-binding module[29,43] and binds to the proximal face of the ring-shaped Hfq hexamer[29,50]. Because Hfq forms a stable 1:1 complex with SgrS[27], a single Hfq hexamer must bind to the poly-U tail, both of the stem-loops, and the UA-rich region upstream of the small stem-loop simultaneously[27,51,60–62]. Hfq brings about a distortion in the mRNA structure, promoting the base-pairing between the RNAs[63,64]. In this study, we showed that the U224 mutations in the poly-U tail caused $k_{on}$ of the RNAs to decrease and $k_{off}$ to increase. Because U224 is distant from the mRNA annealing region of SgrS, our data showed that Hfq directly facilitates RNA–RNA annealing in vivo. The same effect was observed from other mutants that disrupt Hfq binding without changing the mRNA annealing region of SgrS, and collectively our work presents an in vivo evidence that Hfq directly facilitates target binding. It should be emphasized that a careful accounting of SgrS mutations' effects on SgrS lifetimes was necessary to reach this conclusion. Microscopic mechanisms for Hfq's role in sRNA–mRNA annealing are still a subject of active research[60,65,66], and may be investigated in the future using our analysis platform.

## Methods

**Construction of plasmids for Sort-Seq studies.** A ptsG-sfGFP reporter system was constructed, containing 105 nt 5'-UTR and 30 nt coding sequence of ptsG mRNA, which coded for the first ten amino acids of PtsG protein, and this was fused by a 42 nt linker sequence and the superfolder GFP-coding sequence. The reporter system was subcloned from the pZEMB8 plasmid. A plasmid, pAS06 was constructed by inserting the reporter sequence into the low copy plasmid pAS05 between the XhoI and XbaI restriction sites, and the expression of the reporter system was under the control of P$_{Llac-O1}$.

The SgrS sRNA sequence was inserted in between the NdeI and BamHI restriction sites of the medium copy plasmid pZAMB1, and its expression was under the control of P$_{Ltet-O1}$. The sgrS mutation library was prepared by using the plasmid pZAMB1 as a template for mutagenesis PCR and also as a vector to insert the sgrS mutation sequence.

**Cell culture and induction for Sort-Seq studies.** The E. coli MB1 strain ($\Delta ptsG$, $\Delta sgrS$, lacI$^q$, and tetR) was transformed with plasmids (pAS06 for ptsG-sfGFP and pZAMB1 for sgrS or the SgrS mutation library), and grown at 37 °C in LB Broth Miller (EMD) overnight with the respective antibiotics (100 µg ml$^{-1}$ ampicillin (Gold Biotechnology, Inc.)) for pAS06 plasmid and 30 µg ml$^{-1}$ chloramphenicol (Cm; Sigma-Aldrich) for pZAMB1 plasmid and the sgrS mutation library). The following day, the cell culture was diluted 200-fold into fresh LB Broth with

respective antibiotics and were grown until $OD_{600}$ reached 0.1–0.2 as measured using an Educational Spectrophotometer (Fisher Scientific Education). The culture was diluted again to an $OD_{600}$ of 0.001 and supplemented with 1 mM IPTG (Sigma-Aldrich) to induce the expression of PtsG-sfGFP, and 50 ng ml$^{-1}$ aTc to induce the expression of SgrS or the SgrS mutation library. The *E. coli* cells were collected and treated further for the next set of experiments.

**SgrS sRNA mutagenesis experiment**. Agilent Genemorph II Random Mutagenesis Kit (Agilent Technologies) was used to perform mutagenesis PCR on SgrS using the protocol adapted from previously published work from Levine's lab[39], and the steps of the method are as follows. A total of 1 ng of pZAMB1 plasmid, with the *sgrS* sequence, was used to conduct mutagenesis PCR for 15 cycles. The yields of the individual mutants were increased by amplifying the product using Phusion® High-Fidelity DNA Polymerase (New England Biolabs) and purified using QIAquick Spin Columns (Qiagen). The PCR products were then digested with NdeI (New England Biolabs) and BamHI (New England Biolabs), and purified by QIAquick Spin Columns (Qiagen). The pZAMB1 vector was also prepared by digestion with NdeI and BamHI followed by purification using QIAquick Spin Columns (Qiagen). The vector and the PCR insert were used to prepare four ligation reactions by mixing with T4 Ligase (New England Biolabs).

The products from all the reactions were combined and purified using QIAquick Spin Columns (Qiagen) into water. A total of 5 μl of the purified ligation product was then transformed into MB1 strain, which was pre-transformed with pAS06 plasmid expressing ptsG-sfGFP. These transformed cells were then recovered and diluted into LB Broth supplemented with 100 μg ml$^{-1}$ ampicillin and 30 μg ml$^{-1}$ Cm, and grown overnight at 37 °C. The following day, the culture was centrifuged, aliquoted as frozen stocks, and used for imaging and flow cytometry experiments.

**Epifluorescence Imaging**. A total of 1 ml culture of the *E. coli* strain to be imaged was grown from an overnight culture till $OD_{600}$ = 0.1–0.2. It was then chilled on ice followed by centrifugation at $6000 \times g$, 4 °C for 1 min to form a cell pellet. Then they were washed with ice-cold 1× PBS twice and resuspended in 100 μl 1× PBS.

In all, 1.5% (w/v) agarose gel was prepared by dissolving agarose in 1× PBS. A few μl cell suspension was sandwiched between a No. 1.5 glass coverslip (VWR) and a thin slab of the agarose gel. The sample was then imaged.

The epifluorescence images were acquired by a Nikon Ti Eclipse microscope (Nikon Instruments, Inc.) using an oil immersion objective (1.46 NA, 100×), which spans an area of ~133 × 133 μm$^2$ for DIC (no filter, autofluorescence) and fluorescence imaging (Ex 480–500 nm, Em 509–547 nm, exposure time 200 ms). The images were acquired using an EMCCD camera (Andor). They were processed using the NIS-Element AR software (Nikon Instruments, Inc.).

**Fluorescence-activated cell sorting**. The *E. coli* strain to be sorted was cultured overnight in LB Broth with appropriate antibiotics. The following day, the liquid culture was diluted 200-fold and cultured with antibiotics until $OD_{600}$ = 0.1–0.2. The cells were then diluted to $OD_{600}$ = 0.001 in LB Broth with antibiotics and 1 mM IPTG and/or 50 ng ml$^{-1}$ aTc were added corresponding to the strain of *E. coli* and the plasmids it is carrying. They were grown till $OD_{600}$ = 0.1–0.2, washed with ice-cold 1× PBS twice and kept on ice before flow cytometry analysis or fluorescence-activated cell sorting (FACS). The sorting and analysis were done in a MoFlo XDP Cell Sorter (Beckman Coulter) using a 488 nm 200 mW laser.

**Preparation of the sample for sequencing**. The cells sorted into the batches were grown in LB Broth supplemented with 30 μg ml$^{-1}$ Cm to saturation. We extracted the plasmids with E.Z.N.A. Plasmid Mini kit (Omega, D6942-02). To generate sequencing amplicons, we followed Illumina 16S sequencing protocol. We used 5 ng of each plasmid elute as PCR template and amplified out the portion of interest using 0.5 μM of primers annealing to the region of the *sgrS* sequence under consideration with Phusion 2× Mastermix (NEB, M0531L). We employed 20 cycles of 10 s at 98 °C denaturation, 20 s at 63 °C primer annealing, 10 s at 72 °C elongation phases preceded by additional initial denaturation at 98 °C for 30 s, and followed by 72 °C final extension for 2 min. To clean up the product, we incubated the PCR product with 20 μl Ampure-XP beads (Beckman Coulter, A63880) for 5 min. We retained the bead-bound material after keeping for 2 min on a magnetic rack (GE, 1201Q46). We washed the beads twice with 80% ethanol, air-dried for 10 min, and eluted the material in 53 μl 10 mM Tris pH 8.5 by incubation for 2 min. We collected 45–50 μl bead-free liquid 2 min after placing the material on a magnetic rack.

**Illumina next-gen sequencing**. We performed eight additional cycles of PCR with Nextera 24-Index kit for indexing before sample pooling (Illumina, FC-121-1011), for which we used 7.5 μl of the above elute as template, 7.5 μl each of the suitable i5 and i7 primers with 38 μl Phusion 2× Mastermix. We followed manufacturer's recommended thermal cycling protocol (95 °C 3 min, 98 °C 30 s, 55 °C 30 s, 72 °C 30 s, and 72 °C 5 min). We also bead-purified 55 μl of this final product with 56 μl Ampure-XP beads and eluted with 28 μl 10 mM Tris pH 8.5 buffer. We pooled the final products based on their Nanodrop reading at equal molar stoichiometry and diluted the sample down to 4 nM in 10 mM Tris pH 8.5 buffer. We alkaline-denatured by mixing 5 μl DNA sample with 0.2 M NaOH and incubating for 5 min at room temperature. We diluted this product down to 20 pM in Hbf buffer. We loaded a final mixture of 465 μl Hbf buffer, 120 μl pooled 20 pM library, and 15 μl denatured 20 pM PhiX control library (Illumina, FC-110-3001) after a 2 min heat treatment at 96 °C followed by a 5 min incubation on ice. We used a 150 cycles MiSeq v3 reagent kit (Illumina, MS-102-3001) to perform a single-end sequencing for 150 cycles. We used the manufacturer's default algorithm for base calling and de-multiplexing of the constituent samples.

**Intensity moment calculation**. We parsed the raw.fastq output files via a simple home-made C++ script compiled with GCC v. 7.5 and plotted with GNU Octave v. 4.2. The analysis scripts can be accessed via the Gitlab page https://gitlab.com/tuncK/sortseq/-/tree/master and the raw data can be obtained from TH upon request.

We only imported the base calls of each read, thus including all sequences regardless of their quality factors. We directly extracted from each read the subsequence excluding the PCR adaptors, i.e., bases 23–128. Out of this list of subsequences, we detected ones that are exact duplicates of each other by building a red-black binary search tree. Among all such groups, we only considered SgrS sequence variants that are represented by at least ten distinct reads in the data. We compared the observed sequence of each group with the wild-type SgrS sequence, i.e., that of the plasmid used as error-prone PCR template.

We normalized the raw number of reads of each group by both the total number of reads and the fraction of cells falling under each gate. As such, we defined a weighted average intensity to each individual mutant along this 106 base long SgrS segment that we probed. Referred to as the "intensity moment" from now on, we calculated the following quantity:

$$K_{ij} = \frac{\sum_{k=-2}^{2} I_k c^k n_{ij}^k / N^k}{\sum_{k=-2}^{2} c^k n_{ij}^k / N^k} \qquad (1)$$

where, $K_{ij}$ is the intensity moment of the mutant carrying a single substitution mutation at the $i$th base position to nucleotide type $j$ rather than the wt base. $c^k$ is the overall fraction of cells that are sorted into the $k$th bin based on the GFP intensity histogram that FACS acquisition software reports. $n_{ij}^k$ is the number of reads carrying a single substitution mutation at base position $i$ to base type $j$ and detected in the $k$th FACS bin. $N^k$ is the total number of acceptable reads in the dataset. $N^k \geq \sum_{ij} n_{ij}^k$ due to experimental errors, as well as reads carrying multiple substitutions due to the stochastic nature of error-prone PCR. $I_k$ is the median intensity of the cells falling into the $k$th bin as reported by FACS. For the representative intensity of each bin, we used the median intensity reported by the FACS device.

In the figures, we reported the standard score of each entry given by

$$z = \frac{K_{ij} - \langle K_{ij} \rangle}{\sigma_{ij}(K_{ij})} \qquad (2)$$

**Construction of bacterial strains**. The oligonucleotides, plasmids, and strains used in this study are listed in Supplementary Tables 1 and 2. *E. coli* K12 MG1655 derivatives were used for all experiments. P1 transduction[67] or λ-red recombination[68] were used to move alleles between strains. DNA fragments were PCR amplified using Q5® Hot Start High-Fidelity 2× Master Mix (NEB) and oligonucleotides described in Supplementary Table 2. A set of plasmids (Supplementary Table 1) were used as templates to PCR amplify the wild-type and *sgrS* mutants A177T, G178T, G178A, and G184A, using single-stranded oligos (Supplementary Table 2) containing 5′ and 3′ homology to the flanking regions of *cat-sacB* cassette (MB205). DNA fragments containing the *sgrS* mutants G215A, T224G, T224A, T181A, and T182A were PCR amplified from MG1655 genomic DNA, using oligonucleotides listed in Supplementary Table 2.

Translational *ptsG'-'lacZ* reporter fusion under the control of the $P_{BAD}$ promoter (strain MB130) was constructed by PCR amplifying fragment of interest using primer pair MBP201F/MBP201R containing 5′ homologies to $P_{BAD}$ and *lacZ*. PCR product was recombined into *E. coli* PM1205 using λ Red-mediated homologous recombination and counter-selection against *sacB*, as described previously[69]. Marked *λattB::lacI$^q$-PN25tetR-spec$^R$* was introduced into MB130 strain by P1 vir transduction[67] to produce MB168 strain (Supplementary Table 1). Plasmid pZAMB1 harboring *sgrS* under the control of the $P_{LtetO1}$ promoter was constructed by PCR amplifying *sgrS* from *E. coli* MG1655 chromosomal DNA, using oligonucleotides containing NdeI and BamHI restriction sites. PCR products and vector pZA31R[70] were digested with NdeI and BamHI (New England Biolabs) restriction endonucleases. Digestion products were ligated using DNA Ligase (New England Biolabs) to produce plasmid containing $P_{LtetO1}$-*sgrS* allele[40]. Single-nucleotide mutations in SgrS were introduced by QuikChange mutagenesis procedure, using oligonucleotides with mismatched bases at desired locations as following: A177T (A177T-F/A177T-R), G178T (G178T-F/G178T-R), G178A (G178A-F/G178A-R), G184A (G184A-F/G184A-R), G215A (G215A-F/G215A-R), T224A (U224A-F/U224A-R), and T224G (T224G-F/T224G-R) (Supplementary Table 2). All bacterial strains and plasmids are available from the corresponding authors upon reasonable request.

**β-galactosidase assay**. Bacterial strains were cultured overnight in MOPS rich medium with 25 µg ml$^{-1}$ Cm, and subcultured 1:100 to fresh MOPS rich medium containing Cm and 0.0005% L-arabinose. Cells were grown at 37 °C with shaking to OD$_{600}$ ~0.15 and 30 ng ml$^{-1}$ aTc was added to induce expression of SgrS from the plasmid and cells grown for another hour to OD$_{600}$ ~0.5. β-galactosidase assay was then performed on four biological replicates according to previously described protocol[67]. In summary, 1.5 ml of each cell culture was incubated on ice for 20 min, then 1 ml was used to measure A600 and 500 µl were transferred to fresh tubes containing 500 µl of the lysis buffer (0.06 M Na$_2$HPO$_4$, 0.04 M NaH$_2$PO$_4$, 0.01 M KCl, 0.001 M MgSO$_4$, 0.05 M β-mercaptoethanol, pH 7.0) and vigorously mixed. Samples were incubated for 5 min at 28 °C and 0.2 ml of 4 mg ml$^{-1}$ o-nitrophenyl-β-D-galactopyranoside (ONPG) in 0.1 M phosphate buffer (0.06 M Na$_2$HPO$_4$, 0.04 M NaH$_2$PO$_4$, pH 7.0) was added to each sample to start the reaction. When samples changed color to yellow the reaction was stopped by the addition of 0.5 ml of 1 M Na$_2$CO$_3$ and time elapsed since the addition of ONPG was recoded. Samples were centrifugated at 16,000 × $g$ and A420 of the supernatant was measured. β-galactosidase activity was calculated according to the formula $(1000 \times A420)/(T \times V \times A600)$, where $T$ is the time of the reaction in minutes and $V$ is the volume of the culture used in ml. Data were obtained from four independent experiments.

**Northern blot analysis**. Bacterial strains were cultured and β-galactosidase assay was performed, as described above. Simultaneously, aliquots of the same culture were taken and total RNA was extracted, as described previously[71]. RNA concentrations were measured spectrophotometrically and 15 µg of RNA was resolved on 6% polyacrylamide gel electrophoresis. RNA was transferred to 0.2 µm pore size Nytran N (Whatman) membrane, as described previously[72]. Membrane was pre-hybridized for 45 min in ULTRAhyb (Ambion) solution at 42 °C. Blots were probed overnight with 5'-biotinylated SgrS-bio or ssrA-bio probes specific for SgrS sRNA and 5 S rRNA, respectively (Supplementary Table 2). BrightStar BioDetect kit (Ambion) was used for detection. ImageJ (National Institutes of Health[73]) was used to measure band densities from two independent experiments.

**Measurement of intrinsic degradation rates of SgrS**. Strain DB166, MB206, MB209, and XM199 were cultured overnight at 37 °C and diluted 1:100 to a fresh LB medium, and the cultures were grown at 37 °C to OD$_{600}$ ~0.3. To induce SgrS expression, αMG was added to final concentration of 0.5% and the cells were grown for additional 30 min. Rifampicin was added to final concentration of 250 µg ml$^{-1}$ and the cells were grown for another 5 min. At this point, cells were harvested ($t =$ 0 time point) for RNA extraction. Three biological replicates were harvested for each time point.

Cells from 1.0 ml of culture were mixed with 2 ml of RNA protect reagent (Qiagen). The mixture was pelleted at 4000 r.p.m. for 10 min and then discard the supernatant. Total RNA was isolated using Direct-Zol RNA miniPrep (Zymo) kit following the manufacturer's instruction. Genomic DNA was removed by DNaseI provided by the Kit. Finally total RNA was eluted in 40 µl of nuclease-free water. First-strand cDNA was synthesized from 1 µg of total RNA using Superscript™ IV First-Strand cDNA Synthesis SuperMix kit according to the manufacturer's protocol (Invitrogen, USA).

The primers used to amplify SgrS are: OSA499 (GATGAAGCAAGGGGGTG CCC) and OSA500 (CAATACTCAGTCACACATGATGCAGGC).

The primers used to amplify housekeeping gene *rrsA* are: OXM187 (ATTC CGATTAACGCTTGCAC) and OXM188 (AGGGCCTTCGGGTTGTAAAGT).

Real-time PCR was performed using SYBR Green master mix (Fisher) and Eppendorf Realplex in a 96-well plate. Each reaction is comprised of 1× SYBR Green master mix, 100 nM of each primer, and 2 µl of 1:50 diluted cDNA in a total of 10 µl reaction volumes. Each plate contains "no template" controls for individual transcripts, as well as housekeeping transcripts, such as *rrsA* for every sample as an internal control.

Delta delta Ct method was used to analyze the qPCR data. The transcripts turnover rates were calculated based on the nonlinear fit with one-phase exponential decay curves using GraphPad software.

**Cell culture, fixation, and permeabilization for smFISH and super-resolution imaging**. The wild-type *E. coli* strain (DJ480) was grown overnight at 37 °C, 250 r.p. m. in LB Broth Miller (EMD), the RNase E mutant was grown in 25 µg ml$^{-1}$ kanamycin (Kan; Fisher Scientific), the SgrS A177U, G178U, G178A, U181A, U182A, G184A, G184A–C195U, G215A, U224A, and U224G mutants were grown in LB Broth with 50 µg ml$^{-1}$ spectinomycin (Spec; Sigma-Aldrich), and the RNase E mutants of the respective SgrS mutations were grown in LB Broth with 25 µg ml$^{-1}$ Kan and 50 µg ml$^{-1}$ Spec. The following day, the overnight cultures were diluted 100-fold into MOPS EZ rich defined medium (Teknova) with 0.2% glucose and the respective antibiotics, and allowed to grow at 37 °C and 250 r.p.m. until the OD$_{600}$ reached 0.15–0.25. α-methylglucoside (αMG; Sigma-Aldrich) was used to introduce sugar-phosphate stress and subsequently induce SgrS sRNA expression. A specific volume of liquid was taken out of the culture after 0, 2, 4, 6, 8, 10, 15, and 20 min of incubation, and mixed with formaldehyde (Fisher Scientific) to a final concentration of 4% for the fixation of the cells.

Δ*sgrS* and Δ*ptsG* strains were grown overnight in LB Broth Miller (EMD) at 37 °C and 250 r.p.m., using 25 µg ml$^{-1}$ Kan and 10 µg ml$^{-1}$ tetracycline (Sigma-Aldrich), respectively. The next day the cultures were diluted 100-fold into MOPS EZ rich defined medium (Teknova) with 0.2% glucose (Sigma-Aldrich) and the respective antibiotics, and left to grow at 37 °C and 250 r.p.m. again till the OD$_{600}$ reached 0.2. The cells were then mixed with formaldehyde (Fisher Scientific) to a final concentration of 4% to fix the cells.

Following the formaldehyde fixation, the cells were incubated at room temperature for 30 min and subsequently centrifuged at 3214 × $g$ for 10 min at room temperature. The pellets were resuspended in 200 µl 1× PBS and then washed three times, each time performing centrifugation at 600 × $g$ for 4 min and resuspending in 200 µl 1× PBS. The cells were then permeabilized with 70% ethanol, shaken at room temperature for 1 h, and stored at 4 °C prior to fluorescence in situ hybridization.

**Single-molecule fluorescence in situ hybridization**. Stellaris Probe Designer was used to design the smFISH probes and they were ordered from Biosearch Technologies (https://www.biosearchtech.com/). The probe labeling was performed by using equal volumes of each probe. The final volume of sodium bicarbonate was adjusted to 0.1 M by adding 1/9 reaction volume of 1 M sodium bicarbonate (pH = 8.5). 0.05–0.25 mg of Alexa Fluor 647 succinimidyl ester (Life Technologies) or CF 568 succinimidyl ester (Biotium) dissolved in 5 µl DMSO was mixed with the probe solution. The dyes were kept at a molar excess of 20–25-fold relative to the probes. The reaction mixture was incubated in the dark at 37 °C with gentle vortexing overnight. The following day the reaction was quenched by using 1/9 reaction volume of 3 M sodium acetate (pH = 5). Ethanol precipitation followed by P-6 Micro Bio-Spin Columns (Bio-Rad) were employed to remove unconjugated dyes.

A total of 60 µl of permeabilized cells were centrifuged at 600 × $g$ for 4 min and the pellets were washed with FISH wash solution (10% formamide in 2× saline sodium citrate (SSC) buffer). They were then resuspended along with the probes in 15 µl of FISH hybridization buffer (10% dextran sulfate (Sigma-Aldrich), 1 mg ml$^{-1}$ *E. coli* tRNA (Sigma-Aldrich), 0.2 mg ml$^{-1}$ bovine serum albumin (NEB), 2 mM vanadyl ribonucleoside complexes (Sigma-Aldrich), 10% formamide (Fisher Scientific) in 2× SSC). The number of probes used for sRNA SgrS was 9, they were labeled with Alexa Fluor 647 and the concentration of the labeled probes was 50 nM. The number of probes used for *ptsG* mRNA was 28, they were labeled with CF 568 and the labeled probe concentration was 15 nM. The reaction mixtures were incubated in the dark at 30 °C overnight. The following day, the cells were suspended in 20× volume FISH wash solution and centrifuged. They were resuspended in FISH wash solution, incubated at 30 °C for 30 min and centrifuged, and this was repeated three times. After the final washing step, the cells were pelleted and resuspended in 20 µl 4× SSC and stored at 4 °C prior to imaging.

**Single-molecule localization-based super-resolution imaging**. The labeled cells were immobilized on 1.0 borosilicate chambered coverglass (Thermo Scientific Nunc Lab-Tek) treated with poly-L-lysine (Sigma-Aldrich) and imaged with imaging buffer (50 mM Tris-HCl (pH = 8.0), 10% glucose (Sigma-Aldrich), 1% β-mercaptoethanol (Sigma-Aldrich), 0.5 mg ml$^{-1}$ glucose oxidase (Sigma-Aldrich), and 0.2% catalase (Sigma-Aldrich) in 2× SSC).

3D super-resolution imaging was performed using an Olympus IX-71 inverted microscope with a 100× NA 1.4 SaPo oil immersion objective. Sapphire 568-100 CW CDRH (568 nm; Coherent) and DL-640-100-AL-O (647 nm; Crystalaser) were used for two-color imaging and DL405-025 (405 nm; Crystalaser) was used for the reactivation of the dyes. The laser excitation was controlled by mechanical shutters (LS6T2, Uniblitz). The laser lines were reflected to the objective using a dichroic mirror (Di01-R405/488/561/635, Semrock). The emission signal was collected by the objective and then it passed through an emission filter (FF01-594/730-25, Semrock for Alexa Fluor 647 or HQ585/70 M 63061, Chroma for CF 568), and the excitation laser was cleaned using notch filters (ZET647NF, Chroma; NF01-568/647-25 × 5.0, Semrock and NF01-568U-25, Semrock). The images were captured on a 512 × 512 Andor EMCCD camera (DV887ECS-BV, Andor Tech). 3D imaging was achieved by introducing astigmatism using a cylindrical lens with focal length 2 m (SCX-50.8-1000.0-UV-SLMF-520-820, CVI Melles Griot) in the emission path between two relay lenses of focal lengths 100 and 150 mm. Each pixel corresponded to 100 nm in this setup. The z-drift of the setup was controlled by the CRISP (Continuous Reflective Interface Sample Placement) system (ASI) and the region of interest for imaging was selected using an xy-sample stage (BioPrecision2, Ludl Electronic Products). The storm-control software written in Python by Zhuang's group and available at GitHub (https://github.com/ZhuangLab/storm-control) was used for image acquisition.

After acquiring a DIC image of the sample area, two-color super-resolution imaging was performed. A laser excitation of 568 nm was used for CF 568 after completing the image acquisition for Alexa Fluor 647 using 647 nm laser excitation. Fluorophore bleaching was compensated and moderate signal density was maintained by increasing the 405 nm laser power slowly. Imaging was completed when most of the fluorophores had photobleached and the highest reactivation laser power was reached.

Fluorescent nanodiamonds (140 nm diameter, Sigma-Aldrich) were utilized for mapping of the two channels. These nanodiamonds nonspecifically attached to the surface of the imaging chambers, and were excited by both 647 and 568 nm lasers. They generated localization spots in the final reconstructed images that was used for mapping.

**Image analysis**. The raw data was acquired using the Python-based acquisition software and it was analyzed using a data analysis algorithm, which was based on work published previously by Zhuang's group[74,75]. The peak identification and fitting were performed using the method described before and it involved Gaussian blurring, calculation of local maximum intensity pixels in a 5 × 5 pixel area, addition of sharpness and roundness filters and fitting with an Elliptial Gaussian function[34]. The z-stabilization was done by the CRISP system and the horizontal drift was calculated using fast Fourier transformation on the reconstructed images of subsets of the super-resolution image, comparing the center of the transformed images and corrected using linear interpolation.

**Clustering analysis and copy number calculation**. A density-based clustering analysis algorithm (DBSCAN) was employed to calculate the RNA copy numbers. The algorithm used involved clustering analysis, baseline correction, and analysis using Bernoulli trials and was the same as previously published[34], but the Nps and Eps values were updated for the SgrS and *ptsG* images, since we used CF 568 instead of Alexa Fluor 568 and we also used a different 405 nm laser to reactivate the dyes. The SgrS (9 probes labeled with Alexa Fluor 647) images were clustered using Nps = 3 and Eps = 15, and the *ptsG* (28 probes labeled with CF 568) images were clustered using Nps = 10 and Eps = 25, and these numbers were empirically chosen. A MATLAB code was used as before for the cluster analysis.

*ΔsgrS* and *ΔptsG* strains were grown, prepared, imaged, and analyzed in the same manner as before, and they were used for the measurement of the background signal due to the nonspecific binding of Alexa Fluor 647 and CF 568.

The SgrS image with no αMG induction for the wild-type *E. coli* cells (DJ480) was considered to be the low SgrS copy number sample, where it was assumed that one cluster was equivalent to one RNA and the *ptsG* image with 20 min αMG induction for the wild-type *E. coli* cells was considered to be the low *ptsG* copy number sample. The copy numbers of the RNAs were calculated in the same manner using MATLAB codes after clustering analysis, baseline correction, and analysis using Bernoulli trials[34].

**Colocalization analysis**. To calculate the copy number of SgrS-*ptsG* complexes, colocalization analysis was performed in order to calculate the percentage of *ptsG* colocalized with SgrS. The average radius of a *ptsG* mRNA cluster was calculated to be ~40 nm. That value was used as the radius to consider a 3D spherical volume from the center of the *ptsG* cluster. The SgrS spots corresponding to clusters found in this volume were taken to be colocalized with the *ptsG* cluster. The base-pairing mutant strain was considered a negative control (Supplementary Fig. 41a) and percentage of colocalization was plotted against SgrS copy number and fit with a line (y = a × x) to act as a calibration for colocalization by chance (Supplementary Fig. 41b). The coefficient, a, was used as correction factor for colocalization calculation as, final colocalization = calculated colocalization − a × SgrS copy number.

**SgrS and *ptsG* mRNA half-life measurements**. The *ptsG* mRNA degradation rates were calculated using a rifampicin-chase experiment. The wild-type (DJ480) *E. coli* cells, the SgrS A177U, G178A, G178U, U181A, U182A, G184A, G184A–C195U, G215A, U224A, and U224G were grown in LB Broth with the respective antibiotics at 37 °C, 250 r.p.m. overnight. The following day, the overnight cultures were diluted 100-fold in MOPS EZ rich defined medium supplemented with 0.2% glucose and they were grown at 37 °C, 250 r.p.m. When the OD$_{600}$ reached 0.15–0.25 rifampicin (Sigma-Aldrich) was added to a final concentration of 500 μg ml$^{-1}$. This was taken as the 0-min time point for the experiment and aliquots were taken at 2, 4, 6, 8, 10, 15, and 20 min after the addition of rifampicin, and fixed in the same manner described before. The cells were labeled by FISH probes, imaged, and analyzed by the same process mentioned. The natural logs of the copy numbers were plotted against time and the slope of the linear fitting was used to calculate the lifetime of the RNA. The reciprocal of the lifetimes gave the degradation rates.

The SgrS degradation rates were calculated for the above strains and the wild-type Δ*hfq*, A177U Δ*hfq*, and G184A Δ*hfq* mutants by stopping the transcription of SgrS by removing αMG from the media. The wild-type *E. coli* cells, the mutants, and the RNase E mutants were grown overnight, as described before in LB Broth with the respective antibiotics. The cells were diluted the following day and grown in MOPS EZ rich defined medium with the respective antibiotics till OD$_{600}$ 0.15–0.25. SgrS transcription was induced in the cells, using αMG and growing them for 10 min. The cells were then washed twice with centrifugation and resuspension with cold, fresh media devoid of αMG, and finally resuspended in pre-warmed media at 37 °C. Aliquots were taken at 0, 2, 4, 6, 8, 10, 15, and 20 min (0, 2, 4, 6, and 8 min for the Δ*hfq* strains) and fixed, as described before. The cells were then treated, imaged, and analyzed to calculate the degradation rates as

mentioned before.

**Modeling of SgrS-induced *ptsG* mRNA degradation**
*Kinetic model and experimental measurements of the parameters*. The mass-action equations used for the wild-type *E. coli* cells and the chromosomal mutations are shown below:

$$\frac{d[p]}{dt} = \alpha_p - \beta_p[p] - k_{on}[S][p] + k_{off}[Sp] \tag{3}$$

$$\frac{d[S]}{dt} = \alpha_S - \beta_{S,p}[S] - k_{on}[S][p] + k_{off}[Sp] \tag{4}$$

$$\frac{d[Sp]}{dt} = k_{on}[S][p] - k_{off}[Sp] - k_{cat}[Sp] \tag{5}$$

In the above equations, the changes in the concentration of *ptsG*, SgrS, and the SgrS-*ptsG* complex over time are shown. $\alpha_p$, $\alpha_S$ are the transcription rates of the *ptsG* mRNA and SgrS, respectively; $\beta_p$, $\beta_{S,p}$ are respectively the endogenous degradation rate of ptsG mRNA and the degradation rate of SgrS excluding the co-degradation with *ptsG* mRNA; $k_{on}$, $k_{off}$ are the rates of association and dissociation of SgrS and *ptsG* mRNA, and $k_{cat}$ is the RNase E-mediated co-degradation of SgrS-*ptsG* complex.

We calculated the endogenous degradation rate of *ptsG* mRNA ($\beta_p$) of the wild-type *E. coli*, chromosomal mutations, and the RNase E mutants from the super-resolution imaging and analysis. The degradation rates of SgrS for the cells were calculated by stopping the transcription of SgrS, but this method takes into account target-dependent and target-independent degradation ($\beta_{S, total}$). We also calculated the degradation rate for the respective RNase E mutant strains, and this measurement gave us target-independent degradation and other RNase E-independent degradation ($\beta_{S0}$). These two values provided a higher and lower bound for the endogenous degradation rate of SgrS ($\beta_{S,p}$).

The transcription rate of *ptsG* mRNA was calculated using $\alpha_p = \beta_p \times [p]_0$ and in this equation $[p]_0$ is the concentration of *ptsG* mRNA before the induction of sugar stress in all of the cases. This was done because it was observed previously[34] that the *ptsG* mRNA reached an equilibrium in the cells without SgrS-induced degradation. We calculated this for all the cases, viz., wild-type *E. coli*, SgrS mutants, and the RNase E mutants and the transcription rate of *ptsG* mRNA did not show any significant change.

RNase E mutant cells are not able to degrade SgrS-*ptsG* complex efficiently, but it is a possibility that the complex can degrade endogenously or via other minor degradation pathways. We kept $k_{cat}$ as a fitting parameter and used the measured parameters, $\alpha_p$, $\beta_p$, $\beta_{S, total}$, and $\beta_{S0}$ and the above equations to fit the time courses for all the strains to estimate the five parameters; $\alpha_S$, $\beta_{S,p}$, $k_{on}$, $k_{off}$, and $k_{cat}$.

*Parameter search*. Poisson weighting (total sum of the squares, $SS_{tot} = \Sigma_i(y_i - \bar{y})^2$ and residual sum of the squares, $SS_{res} = \Sigma_i(y_i - f_i)^2$, where $y_i$ is the experimental data and $f_i$ is the fitted data) was used in the fitting of global $R^2$ according to the equation:

$$R^2 \equiv 1 - \frac{SS_{res}}{SS_{tot}} \tag{6}$$

so that no bias was introduced for a particular species. The parameters were selected to maximize the global $R^2$ for the time course curves of each of the species. The concentrations of the SgrS-*ptsG* complex in all the strains were very close to the background and as a result the total variance became small. $R^2$ was not helpful to estimate the quality of the fit in these cases. Instead, $\chi^2$'s were calculated as

$$\chi^2 \equiv \sum_i \frac{(y_i - f_i)^2}{f_i} \tag{7}$$

for all those cases and the significance levels ($\alpha$) were reported.

**Reporting summary**. Further information on research design is available in the Nature Research Reporting Summary linked to this article.

## Data availability
The sequencing data generated for this study are available at the BioProject Database (ID PRJNA666229). The data supporting the findings of this study are available from the corresponding authors upon reasonable request. Source data are provided with this paper.

## Code availability
The Sort-Seq data analysis scripts can be accessed via the Gitlab page (https://gitlab.com/tuncK/sortseq/-/tree/master). The STORM data acquisition code is available at Github (https://github.com/ZhuangLab/storm-control). All other codes used in this study are available from the corresponding authors upon reasonable request.

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

## Acknowledgements

We would like to thank Erel Levine, Divya Balasubramanian, and D. Jin for plasmids and strains. We thank Hao Zhang at the Cell Sorting Core Facility (Bloomberg School of Public Health) for helping us with the flow cytometry and sorting experiments. We appreciate and thank Prof. Sarah Woodson for going through the manuscript and providing insightful suggestions. This work was supported by grants from National Institutes of Health R01 GM112659 (M.B., M.S.A., T.H., J.Z., and A.P.), R35 GM122569 (T.H., J.Z., and A.P.), National Science Foundation PHY 1430124 (T.H., Z.L.-S., J.Z., and A.P). T.H. is an investigator with the Howard Hughes Medical Institute.

## Author contributions

A.P., C.K.V., and T.H. designed the experiments, with help from M.B. and J.Z.; A.P., T.K., and J.Z. performed the Sort-Seq experiments and the Sort-Seq data were analyzed by A.P., T.K., and P.L.; M.B. performed the β-galactosidase and northern blot experiments. M.S.A. and X.M. made the strains that were used in this work. M.S.A. and X.M. performed the qPCR experiments to calculate RNA lifetimes. A.P. performed all super-resolution imaging experiments, with some help from J.Z.; A.P. performed the analysis for the imaging experiments with the MATLAB package written by D.S. and J.F.; A.P., Z.L.-S., C.K.V., and T.H. discussed the data. A.P., T.K., M.S.A., X.M., J.F., C.K.V., and T.H. wrote the manuscript.

## Competing interests

The authors declare no competing interests.
