## [Peer Review File · Nature Communications]

REVIEWER COMMENTS

Reviewer #1 (Remarks to the Author):

The paper by Poddar et al. describes the impact of single nucleotide mutation in Sars sRNA on its kinetic interactions with the ptsG mRNA and Hfq. The in vivo kinetic assay is the same as the group previously has used for the wt sRNA.

In addition they use sort-Seq to to limit the number of positions to mutate to those that impacts regulation.

Overall I think its a very good paper. Everything is clearly described, the logic in reasoning is good and the the data analysis seems very thorough.

The only thing that is somewhat concerning is the failure to predict the sort-seq results from the model, which makes me wonder if the sort-seq strategy worked to pick out the best positions to mutate. However, the authors give reasonable explanations for what may be the problem of predicting the sort-see results. I think it's OK.

A few minor things:

line 93:

"Its non-zero value showed that even for the correct target, binding is reversible. "
This is not a very informative sentence.

line 94:

"The large dissociation constant K_D ($=k_{off}/k_{on}$) of $\sim 1 \mu M$ explained why more than a hundred SgrS molecules are needed for rapid ptsG mRNA regulation"

The equilibrium constant does not say anything about what is needed for a rapid response.

line 204

" The accumulation of the mutant SgrS sRNAs was lower than for wild-type SgrS"

I think the word accumulation is not very good as it is used in this sentence and in a few other palaces. For me accumulation is a statement about rate of change and not amount, as I think is intended.

Reviewer #2 (Remarks to the Author):

In this interesting study, Poddar et al. establish a sorting and sequencing (Sort-Seq) approach to identify SgrS small RNA mutants that are defective for repressing a ptsG-sfGPF fusion. The authors then select 9 single point mutants and 1 double mutant, which they express from the endogenous chromosomal location, to examine the effects of these mutations on SgrS base pairing with the ptsG mRNA in vivo using super-resolution imaging of SgrS and ptsG labeled with FISH probes. Based on their measurements and calculations of association and dissociation rates, the authors conclude that Hfq directly facilitates sRNA-mRNA annealing and buffers base-pair disruptions.

The manuscript is clearly written and very carefully prepared (with 53! Supplementary Figures). I only have relatively minor comments:

1. I think the statements that the "work presents the first in vivo evidence that Hfq directly facilitates

target binding" is a bit of an overstatement. Other in vivo studies with Hfq mutants, sRNA mutants as well as various global RNA sequencing methods of RNA associations with Hfq have contributed evidence that Hfq facilitates sRNA binding to mRNAs.

2. Lines 136-138: The full sequence of the mutants presented in Figure 2d should be determined. The authors have demonstrated how distal mutations can affect an RNA in unexpected ways.

3. Pages 15-16: Ideally, the authors should also present the effects of the point mutants expressed from the endogenous chromosomal location on PtsG protein levels (this could help explain the difference in outcome between the imaging and Sort-seq experiments).

4. The text of the Discussion could be simplified.

REVIEWER COMMENTS

Reviewer #1 (Remarks to the Author):

The paper by Poddar et al. describes the impact of single nucleotide mutation in Sars sRNA on its kinetic interactions with the ptsG mRNA and Hfq. The in vivo kinetic assay is the same as the group previously has used for the wt sRNA.

In addition they use sort-Seq to limit the number of positions to mutate to those that impacts regulation.

Overall I think its a very good paper. Everything is clearly described, the logic in reasoning is good and the the data analysis seems very thorough.

The only thing that is somewhat concerning is the failure to predict the sort-seq results from the model, which makes me wonder if the sort-seq strategy worked to pick out the best positions to mutate. However, the authors give reasonable explanations for what may be the problem of predicting the sort-see results. I think it's OK.

Response: We thank the reviewer for carefully assessing our manuscript and their compliments.

A few minor things:

line 93:

“Its non-zero value showed that even for the correct target, binding is reversible. ”

This is not a very informative sentence.

Response: We replaced the sentence by the following sentence inserted in the same paragraph.

“Finally, k_{off} and k_{cat} are similar in magnitude, suggesting that sRNA-mRNA complex is almost as likely to fall apart as to lead to co-degradation.”

line 94:

“The large dissociation constant $KD (=k_{off}/k_{on})$ of $\sim 1 \mu M$ explained why more than a hundred SgrS molecules are needed for rapid ptsG mRNA regulation”

The equilibrium constant does not say anything about what is needed for a rapid response.

Response: The text has been edited as follows.

“... explained why more than a hundred SgrS molecules are produced during *ptsG* mRNA regulation.”

line 204

“ The accumulation of the mutant SgrS sRNAs was lower than for wild-type SgrS”

I think the word accumulation is not very good as it is used in this sentence and in a few other places. For me accumulation is a statement about rate of change and not amount, as I think is intended.

Response: The text has been edited to change “the accumulation” to “the copy number”.

Reviewer #2 (Remarks to the Author):

In this interesting study, Poddar et al. establish a sorting and sequencing (Sort-Seq) approach to identify SgrS small RNA mutants that are defective for repressing a *ptsG*-sfGFP fusion. The authors then select 9 single point mutants and 1 double mutant, which they express from the endogenous chromosomal location, to examine the effects of these mutations on SgrS base pairing with the *ptsG* mRNA in vivo using super-resolution imaging of SgrS and *ptsG* labeled with FISH probes. Based on their measurements and calculations of association and dissociation rates, the authors conclude that Hfq directly facilitates sRNA-mRNA annealing and buffers base-pair disruptions.

The manuscript is clearly written and very carefully prepared (with 53! Supplementary Figures).

Response: We thank the reviewer for carefully examining our manuscript and their support.

I only have relatively minor comments:

1. I think the statements that the “work presents the first in vivo evidence that Hfq directly facilitates target binding” is a bit of an overstatement. Other in vivo studies with Hfq mutants, sRNA mutants as well as various global RNA sequencing methods of RNA associations with Hfq have contributed evidence that Hfq facilitates sRNA binding to mRNAs.

Response: The text has been edited. We removed “the first”.

2. Lines 136-138: The full sequence of the mutants presented in Figure 2d should be determined. The authors have demonstrated how distal mutations can affect an RNA in unexpected ways.

Response: Our preliminary experiments showed that mutations in the 5' region of SgrS do not have a measurable effect on the regulation of *PtsG* mRNA. Please see the figure below we present for reviews only. Therefore, we believe that the Sort-seq analysis we performed for the manuscript captures almost all of the regions relevant to the regulation of *PtsG* mRNA.

3. Pages 15-16: Ideally, the authors should also present the effects of the point mutants expressed from the endogenous chromosomal location on PtsG protein levels (this could help explain the difference in outcome between the imaging and Sort-seq experiments).

Response: The scope of our study is to examine the mode of regulation that occurs via sRNA-mRNA co-degradation (see schematics in Figure 1). The other mode of regulation, which is translation repression, is another fascinating subject, and will be a subject of future studies and will require fluorescence tagging of PtsG without affecting its function and is presently ongoing in our labs. Regarding the observed differences in imaging and Sort-seq experiments, we used a reporter system where we do not actually express PtsG protein itself. As a result, quantifying endogenously expressed PtsG proteins is not very likely to explain the difference.

4. The text of the Discussion could be simplified.

Response: We have simplified the section.

REVIEWERS' COMMENTS

Reviewer #1 (Remarks to the Author):

I am happy with the minor changes.

Reviewer #2 (Remarks to the Author):

The authors have appropriately addressed the reviewers' comments with edits to their text.

REVIEWERS' COMMENTS

Reviewer #1 (Remarks to the Author):

I am happy with the minor changes.

Response: We thank the reviewer for their time and effort in carefully assessing our manuscript, and for providing us with feedback.

Reviewer #2 (Remarks to the Author):

The authors have appropriately addressed the reviewers' comments with edits to their text.

Response: We thank the reviewer for their time and effort in reviewing the manuscript, for supporting our work and for providing thoughtful comments.